# Diffusion Models for Implicit Image Segmentation Ensembles

**Julia Wolleb**∗                                                  JULIA.WOLLEB@UNIBAS.CH
**Robin Sandkühler**∗                                      ROBIN.SANDKUEHLER@UNIBAS.CH
**Florentin Bieder**                                       FLORENTIN.BIEDER@UNIBAS.CH
**Philippe Valmaggia**                                   PHILIPPE.VALMAGGIA@UNIBAS.CH
**Philippe C. Cattin**                                     PHILIPPE.CATTIN@UNIBAS.CH
*Department of Biomedical Engineering, University of Basel, Allschwil, Switzerland*

## Abstract

Diffusion models have shown impressive performance for generative modelling of images. In this paper, we present a novel semantic segmentation method based on diffusion models. By modifying the training and sampling scheme, we show that diffusion models can perform lesion segmentation of medical images. To generate an image-specific segmentation, we train the model on the ground truth segmentation, and use the image as a prior during training and in every step during the sampling process. With the given stochastic sampling process, we can generate a distribution of segmentation masks. This property allows us to compute pixel-wise uncertainty maps of the segmentation, and allows an implicit ensemble of segmentations that increases the segmentation performance. We evaluate our method on the BRATS2020 dataset for brain tumor segmentation. Compared to state-of-the-art segmentation models, our approach yields good segmentation results and, additionally, detailed uncertainty maps.

**Keywords:** Diffusion models, segmentation, uncertainty estimation

## 1. Introduction

Semantic segmentation is an important and well-explored area in medical image analysis (Rizwan I Haque and Neubert, 2020). The automated segmentation of lesions in medical images with machine learning has shown good performances (Isensee et al., 2021) and is ready for clinical application to support diagnosis (Sharrock et al., 2021). In medical applications, it is of high interest to measure the uncertainty of a given prediction, especially when used for further treatments like radiation therapy.

In this work, we focus on the BRATS2020 brain tumor segmentation challenge (Menze et al., 2014; Bakas et al., 2017, 2018). This dataset provides four different MR sequences for each patient (namely T1-weighted, T2-weighted, FLAIR and T1-weighted with contrast enhancement), as well as the pixel-wise ground truth segmentation. An exemplary image can be found in Appendix A.

We propose a novel segmentation method based on a Denoising Diffusion Probabilistic Model (DDPM) (Ho et al., 2020), which can provide uncertainty maps of the produced segmentation mask. An overview of the workflow for an image of the BRATS2020 dataset is shown in Figure 1. We train a DDPM on the segmentation masks and add the original brain MR image as an image prior to induce the anatomical information. As sampling with

---

∗ Contributed equally

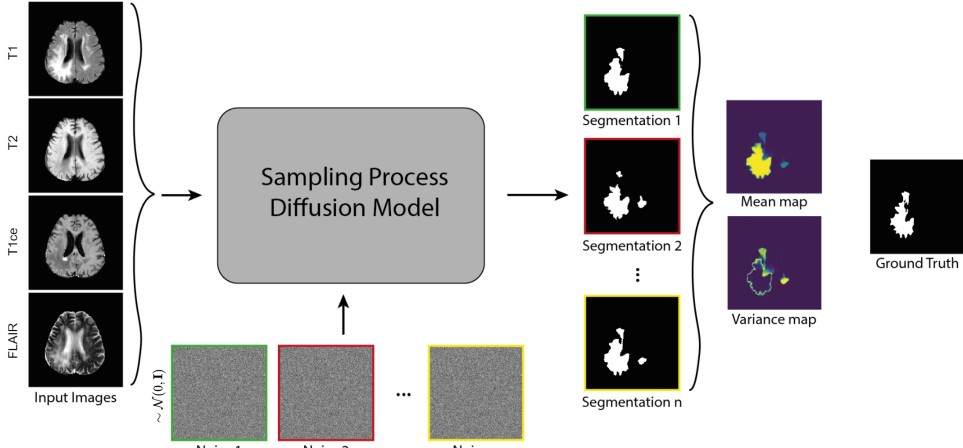

Figure 1: Workflow for the implicit generation of segmentation ensembles and uncertainty maps with diffusion models. The input image consists of four different MR sequences. Going $n$ times through the sampling process of the diffusion model with different Gaussian noise, $n$ different segmentation masks are generated.

DDPMs has a stochastic element in each sampling step, we can generate many different segmentation masks for the same input image and the same pretrained model. This ensemble of segmentations allows us to compute the pixel-wise variance maps, which visualizes the uncertainty of the generated segmentation. Moreover, the ensembling of the segmentations in a mean map boosts the segmentation performance.

We compare ourselves against state-of-the-art segmentation algorithms, and visually compare our variance map against common uncertainty maps. The code is publicly available at https://github.com/JuliaWolleb/Diffusion-based-Segmentation.

**Related Work** In medical image segmentation, a common method is the application of a U-Net (Ronneberger et al., 2015) or SegNet (Badrinarayanan et al., 2017) to predict the segmentation mask for every input image. This approach was successfully applied for many different tasks (Habijan et al., 2019; Kumar et al., 2019; Xiao et al., 2020). The state of the art is given by nnU-Nets (Isensee et al., 2021), where the best architecture and hyperparameters are automatically chosen for every specific dataset.

Uncertainty quantification is of high interest in deep learning research (Abdar et al., 2021), which is often done using Bayesian neural networks (Kendall et al., 2017; Mitros and Mac Namee, 2019; Gal and Ghahramani, 2016). We can differentiate between epistemic uncertainty, which refers to uncertainty in the model parameters, and aleatoric uncertainty, which refers to uncertainty in the data. As stated in (Kendall and Gal, 2017), the epistemic uncertainty of a segmentation model can be approximated with Monte Carlo Dropout, whereas the aleatoric uncertainty can be modeled with Maximum-A-Posteriori inference. Those methods were also applied on various medical tasks (Wang et al., 2019; Nair et al., 2020; DeVries and Taylor, 2018), including brain tumor segmentation (Sagar, 2020; Jungo

and Reyes, 2019; Mehta et al., 2020). Other approaches presented stochastic segmentation networks to model aleatoric uncertainty (Monteiro et al., 2020), or proposed a probabilistic U-Net to learn a distribution over segmentations (Kohl et al., 2018, 2019).

During the last year, DDPMs have gained a lot of attention due to their astonishing performance in image generation (Dhariwal and Nichol, 2021). Images are generated by sampling from Gaussian noise. This sampling scheme follows a stochastic process, and therefore sampling from the same noisy image does not result in the same output image. A different sampling scheme was introduced by Denoising Diffusion Implicit Models (DDIM)(Song et al., 2020), where sampling is deterministic and can be done by skipping multiple steps. Moreover, meaningful interpolation between images can be achieved. DDPM was further improved by (Nichol and Dhariwal, 2021) and (Dhariwal and Nichol, 2021), where changes in the loss objective, architecture improvements, and classifier guidance during sampling improved the output image quality.

While some new work applies diffusion models on tasks such as image-to-image translation (Sasaki et al., 2021), style transfer (Choi et al., 2021), or inpainting tasks (Saharia et al., 2021), so far there is only very little work about semantic segmentation. Recently, one approach to perform semantic segmentation with a diffusion model was proposed by (Baranchuk et al., 2022). A DDPM is trained to reconstruct the image that should be segmented. Then, a multilayer perceptron for classification is applied on the features of the model, which results in a segmentation mask for the original image. In contrast to this method, we train a DDPM directly to generate the segmentation mask. Simultaneously and independent from us, (Amit et al., 2021) developed an image segmentation method similar to ours. However, they use a separate encoder for the image and the segmentation. Training a larger model may be difficult for medical image analysis due to possible large input images such as 3D data. Our method uses only one encoder to encode the image information and the segmentation mask.

## 2. Method

The goal is to train a DDPM to generate segmentation masks. We follow the idea and implementation proposed in (Nichol and Dhariwal, 2021). The core idea of diffusion models is that for many timesteps $T$, noise is added to an image $x$. This results in a series of noisy images $x_0, x_1, ..., x_T$, where the noise level is steadily increased from 0 (no noise) to $T$ (maximum noise). The model follows the architecture of a U-Net and predicts $x_{t-1}$ from $x_t$ for any step $t \in \{1, ..., T\}$. During training, we know the ground truth for $x_{t-1}$, and the model is trained with an MSE loss. During sampling, we start from noise $x_T \sim \mathcal{N}(0, \mathbf{I})$, sample for $T$ steps, until we get a fake image $x_0$.

The complete derivations of the formulas below can be found in (Ho et al., 2020; Nichol and Dhariwal, 2021). The main components of diffusion models are the forward noising process $q$ and the reverse denoising process $p$. Following (Ho et al., 2020), the forward noising process $q$ for a given image $x$ at step $t$ is given by

$$q(x_t|x_{t-1}) := \mathcal{N}(x_t; \sqrt{1 - \beta_t}x_{t-1}, \beta_t\mathbf{I}), \tag{1}$$

where $\mathbf{I}$ denotes the identity matrix and $\beta_1, ..., \beta_T$ are the forward process variances. The idea is that in every step, a small amount of Gaussian noise is added to the image. Doing

this for $t$ steps, we can write

$$q(x_t|x_0) := \mathcal{N}(x_t; \sqrt{\overline{\alpha}_t}x_0, (1 - \overline{\alpha}_t)\mathbf{I}), \tag{2}$$

with $\alpha_t := 1 - \beta_t$ and $\overline{\alpha}_t := \prod_{s=1}^{t} \alpha_s$. With the reparametrization trick, we can directly write $x_t$ as a function of $x_0$:

$$x_t = \sqrt{\overline{\alpha}_t}x_0 + \sqrt{1 - \overline{\alpha}_t}\epsilon, \quad \text{with } \epsilon \sim \mathcal{N}(0, \mathbf{I}). \tag{3}$$

The reverse process $p_\theta$ is learned by the model parameters $\theta$ and is given by

$$p_\theta(x_{t-1}|x_t) := \mathcal{N}\big(x_{t-1}; \mu_\theta(x_t, t), \Sigma_\theta(x_t, t)\big). \tag{4}$$

As shown in (Ho et al., 2020), we can then predict $x_{t-1}$ from $x_t$ with

$$x_{t-1} = \frac{1}{\sqrt{\alpha_t}}\left(x_t - \frac{1 - \alpha_t}{\sqrt{1 - \overline{\alpha}_t}}\epsilon_\theta(x_t, t)\right) + \sigma_t\mathbf{z}, \quad \text{with } \mathbf{z} \sim \mathcal{N}(0, \mathbf{I}), \tag{5}$$

where $\sigma_t$ denotes the variance scheme that can be learned by the model, as proposed in (Nichol and Dhariwal, 2021). We can see in Equation 5 that sampling has a random component $\mathbf{z}$, which leads to a stochastic sampling process. Note that $\epsilon_\theta$ is the U-Net we train, with input $x_t = \sqrt{\overline{\alpha}_t}x_0 + \sqrt{1 - \overline{\alpha}_t}\epsilon$. The noise scheme $\epsilon_\theta(x_t, t)$ that will be subtracted from $x_t$ during sampling according to Equation 5 has to be learned by the model. This U-Net is trained with the loss objectives given in (Nichol and Dhariwal, 2021).

We now modify this idea to use diffusion models for semantic segmentation. A visualization of the workflow is given in Figure 2 for the task of brain tumor segmentation.

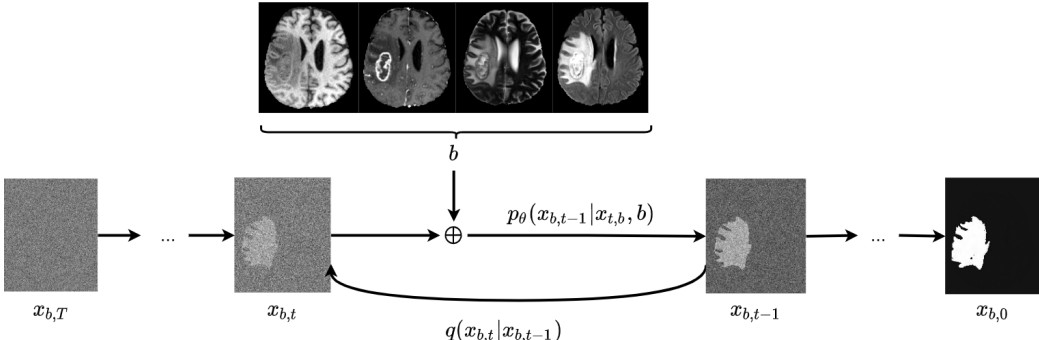

Figure 2: The training and sampling procedure of our method. In every step $t$, the anatomical information is induced by concatenating the brain MR images $b$ to the noisy segmentation mask $x_{b,t}$.

Let $b$ be the given brain MR image of dimension $(c, h, w)$, where $c$ denotes the number of channels, and $(h, w)$ denote the image height and image width. The ground truth segmentation of the tumor for the input image $b$ is denoted as $x_b$, and is of dimension $(1, h, w)$. We

train a DDPM for the generation of segmentation masks. In the classical DDPM approach, $x_b$ would be the only input we need for training, which would result in an arbitrary segmentation mask $x_0$ when we sample from noise during inference. In contrast to that, the goal in our proposed method is not to generate *any* segmentation mask, but we want a meaningful segmentation mask $x_{b,0}$ for a given image $b$. To achieve this, we add additional channels to the input: We induce the anatomical information present in $b$ by adding it as an image prior to $x_b$. We do this by concatenating $b$ and $x_b$, and define $X := b \oplus x_b$. Consequently, $X$ has dimension $(c+1, h, w)$.

During the noising process $q$, we only add noise to the ground truth segmentation $x_b$:

$$x_{b,t} = \sqrt{\overline{\alpha_t}} x_b + \sqrt{1 - \overline{\alpha_t}} \epsilon, \quad \text{with } \epsilon \sim \mathcal{N}(0, \mathbf{I}), \tag{6}$$

and we define $X_t := b \oplus x_{b,t}$. Equation 5 is then altered to

$$x_{b,t-1} = \frac{1}{\sqrt{\alpha_t}} \left( x_{b,t} - \frac{1 - \alpha_t}{\sqrt{1 - \overline{\alpha_t}}} \epsilon_\theta(X_t, t) \right) + \sigma_t \mathbf{z}, \quad \text{with } \mathbf{z} \sim \mathcal{N}(0, \mathbf{I}) \tag{7}$$

and results in a slightly denoised $x_{b,t-1}$ of dimension $(1, h, w)$. During inference, we follow the procedure presented in Algorithm 1, which is a stochastic process. Therefore, sampling twice for the same brain MR image $b$ does not result in the same segmentation mask prediction $x_{b,0}$. Exploiting this property, we can implicitly generate an ensemble of segmentation masks without having to train a new model. This ensemble can then be used to boost the segmentation performance.

---

**Algorithm 1:** Sampling Procedure

---

**Input:** $b$, the original brain MRI
**Output:** $x_{b,0}$, the predicted segmentation mask
sample $x_{b,T} \sim N(0, \mathbf{I})$;
**for** $t \leftarrow T$ **to** 1 **do**
    $X_t \leftarrow b \oplus x_{b,t}$;
    $x_{b,t-1} \leftarrow \frac{1}{\sqrt{\alpha_t}} \left( x_{b,t} - \frac{1-\alpha_t}{\sqrt{1-\overline{\alpha_t}}} \epsilon_\theta(X_t, t) \right) + \sigma_t \mathbf{z}, \quad \text{with } \mathbf{z} \sim \mathcal{N}(0, \mathbf{I})$ ;
**end**

---

## 3. Dataset and Training Details

We evaluate our method on the BRATS2020 dataset. As described in Section 1, images of four different MR sequences are provided for each patient, which are stacked to 4 channels. We slice the 3D MR scans in axial slices. Since tumors rarely occur on the upper or lower part of the brain, we exclude the lowest 80 slices and the uppermost 26 slices. For intensity normalization, we cut the top and bottom one percentile of the pixel intensities. We crop the images to a size of $(4, 224, 224)$. The provided ground truth labels contain four classes, which are background, GD-enhancing tumor, the peritumoral edema, and the necrotic and non-enhancing tumor core. We merge the three different tumor classes into one class and therefore define the segmentation problem as a pixel-wise binary classification. Our training

set includes 16,298 images originating from 332 patients, and the test set comprises 1,082 images with non-empty ground truth segmentations, originating from 37 patients. No data augmentation is applied.

The hyperparameters for our DDPM models are described in the appendix of (Nichol and Dhariwal, 2021). We choose a linear noise schedule for $T = 1000$ steps. The model is trained with the hybrid loss objective, with a learning rate of $10^{-4}$ for the Adam optimizer, and a batch size of 10. The number of channels in the first layer is chosen as 128, and we use one attention head at resolution 16. We train the model for 60,000 iterations on an NVIDIA Quadro RTX 6000 GPU, which takes around one day. The training details for the comparing methods can be found in Appendix B.

## 4. Results and Discussion

During evaluation, we take an image $b$ from the test set, follow Algorithm 1 and produce a segmentation mask. This mask is thresholded at 0.5 to obtain a binary segmentation. In Table 1, the Dice score, the Jaccard index, and the 95 percentile Hausdorff Distance (HD95) are presented. We achieve good results with respect to all those metrics.

For every image of the test set, we sample 5 different segmentation masks. This implicitly defines an ensemble by averaging over the 5 masks and thresholding it at 0.5. We report the results for this ensemble in the second line of Table 1. We see that already an ensemble of 5 increases the performance of our approach.

In the last column of Table 1, we count the cases where the model produces an empty segmentation mask. This results in a Dice of zero, and HD95 cannot be computed. If we disregard those cases, we report the HD95 score, and the average Dice score and Jaccard index are reported in square brackets in Table 1.

As baseline, we report the segmentation scores for the nnU-Net and SegNet. By default, nnU-Net is an ensemble of a 5-fold cross validation. We also implement Bayesian SegNet with Monte Carlo dropout as proposed in (Kendall et al., 2017). By sampling five times during inference, we can again make an ensemble of the generated segmentation masks. The scores for this ensemble are reported in the last line of Table 1.

The generation of one sample with our method takes 48 seconds, while the computation of the segmentation mask with SegNet takes 13 ms. To speed up the sampling process, we will consider sampling with the DDIM approach in future work.

For visualization of the uncertainty maps, we select three exemplary images $b_1$, $b_2$, and $b_3$ from the test set. More examples are presented in Appendix C. To generate detailed

Table 1: Segmentation scores of our method and nnU-Net on different metrics.

| Method | Dice | HD95 | Jaccard | empty |
|---|---|---|---|---|
| Ours (1 sampling run) | 0.866 [0.892] | 6.052 | 0.795 [0.819] | 31 |
| Ours (ensemble of 5 runs) | 0.881 [0.909] | 5.178 | 0.819 [0.845] | 34 |
| nnU-Net (ensemble of 5-fold cross-val.) | 0.891 [0.905] | 5.004 | 0.831 [0.845] | 17 |
| SegNet (1 run) | 0.839 [0.867] | 7.190 | 0.761 [0.786] | 34 |
| Bayesian SegNet (ensemble of 5 runs) | 0.838 [0.841] | 13.707 | 0.747 [0.749] | 3 |

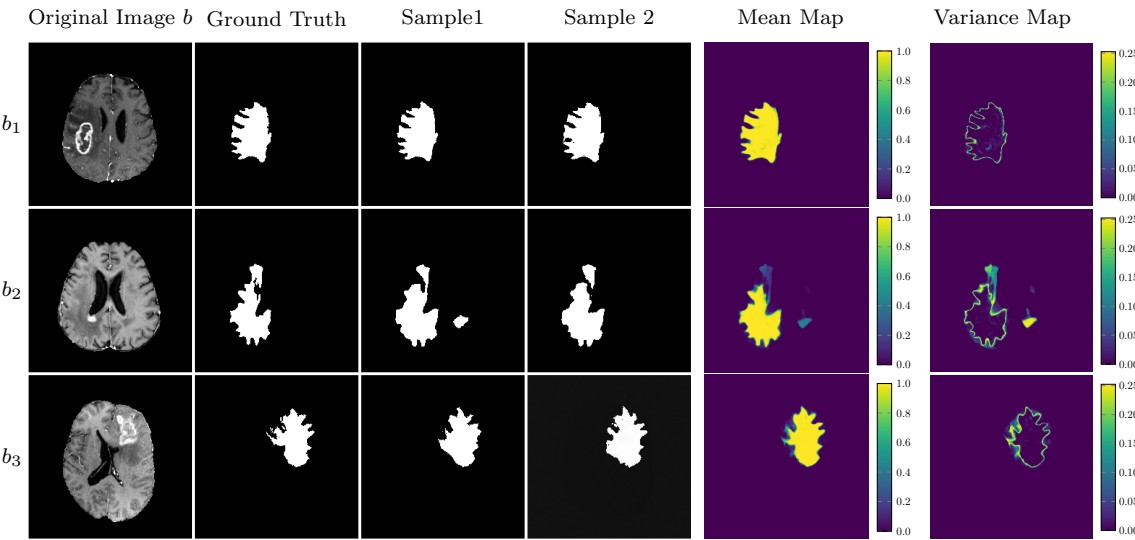

Figure 3: Examples of the produced mean and variance maps for 100 sampling runs.

uncertainty maps, we sample 100 segmentation masks for each of the images, and compute the pixel-wise variance. In Figure 3, we present one channel of the original brain MR image $b$, the ground truth segmentation, two different sampled segmentation masks, as well as the mean and variance map. We can clearly identify the areas where the model was uncertain. Moreover, by thresholding the mean map at 0.5, we can produce the ensembled segmentation mask. In Table 2, we report the segmentation scores and for this ensemble mask, as well as the average scores for the 100 samples. We see that the ensemble can boost the performance for the examples $b_1$, $b_2$ and $b_3$.

Table 2: Segmentation scores for the 100 samples of the examples presented in Figure 3.

| | Average | | | Ensemble | | |
|---|---|---|---|---|---|---|
| Example | Dice | HD95 | Jaccard | Dice | HD95 | Jaccard |
| $b_1$ | 0.969 | 2.360 | 0.939 | 0.981 | 1.000 | 0.962 |
| $b_2$ | 0.869 | 18.503 | 0.769 | 0.885 | 18.468 | 0.783 |
| $b_3$ | 0.932 | 5.227 | 0.872 | 0.952 | 4.474 | 0.907 |

In Figure 4, we plot the number of samples in the ensemble against the Dice score for the three examples $b_1$, $b_2$, and $b_3$. We can see that already an ensemble of five samples improves the performance, and then the curve flattens. In (Amit et al., 2021), a similar experiment was performed on a different data set. Independently from each other, we got the same findings. In Figure 5, we compare our variance maps against the ones of the Bayesian SegNet with Monte Carlo (MC) dropout for 100 samples, as well as the aleatoric uncertainty maps for SegNet, computed as proposed in (Kendall and Gal, 2017).

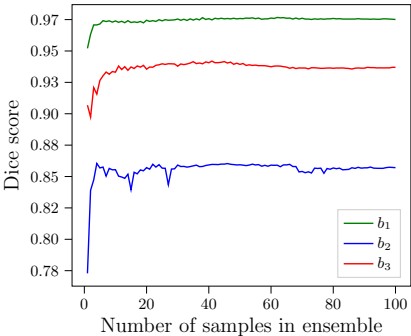

Figure 4: Performance of the ensemble with respect to the number of samples for the examples $b_1$, $b_2$, and $b_3$, presented in Figure 3.

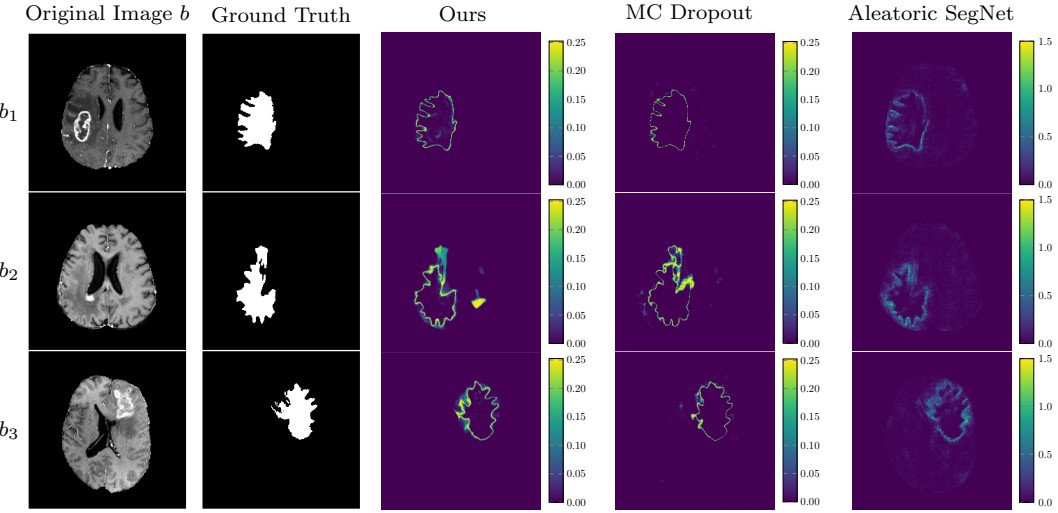

Figure 5: Comparison of the different uncertainty maps for the three examples.

## 5. Conclusion

We presented a novel approach for biomedical image segmentation based on DDPMs. Using the stochastic sampling process, our method allows implicit ensembling of different segmentation masks for the same input brain MR image, without having to train a new model. We could show that ensembling those segmentation masks increases the performance of the model with respect to different segmentation scores. Moreover, we can generate uncertainty maps by computing the variance of the different segmentation masks. This is of great interest in clinical applications, when we want to measure the uncertainty of the decision of the model. For future work, we plan to investigate the segmentation of the different tumor classes provided by the BRATS2020 challenge. Furthermore, we plan to use the DDIM scheme to speed up the sampling process.

## Acknowledgments

This research was supported by the Novartis FreeNovation initiative and the Uniscientia Foundation (project # 147-2018).

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

## Appendix A. Exemplary Image of BRATS2020

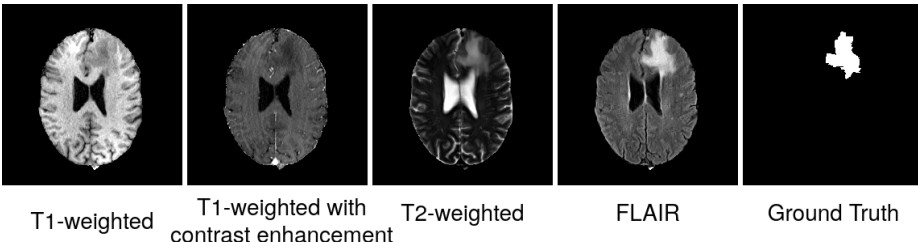

Figure 6: Exemplary image of the BRATS2020 dataset, with four different MR sequences and the ground truth segmentation.

## Appendix B. Implementation Details

We provide implementation details of the comparing methods.

- SegNet: We train the SegNet as proposed in (Badrinarayanan et al., 2017), with a learning rate of $10^{-4}$ for the Adam optimizer and a batch size of 20. Training is performed with the binary cross-entropy loss and is stopped after 100 epochs.

- Bayesian SegNet: We adapt the SegNet architecture, and place the dropout layers with a dropout probability of $p = 0.5$ as proposed in (Kendall et al., 2017). The training schedule is kept the same as for SegNet.

- nnU-Net: We take over all hyperparameter settings as proposed in their official implementation, which can be found at https://github.com/MIC-DKFZ/nnUNet.

- Aleatoric Uncertainty Estimation: We keep the training settings for SegNet. The only change we need to make to the SegNet architecture is to double the number of output channels, such that we get both a prediction and a variance map. We follow the aleatoric loss implementation as proposed in (Jungo and Reyes, 2019), which can be found at https://github.com/alainjungo/reliability-challenges-uncertainty.

## Appendix C. Further Examples

In Figure 7, we provide the mean and variance maps of three more exemplary images $b_4$, $b_5$, and $b_6$ of the test set.

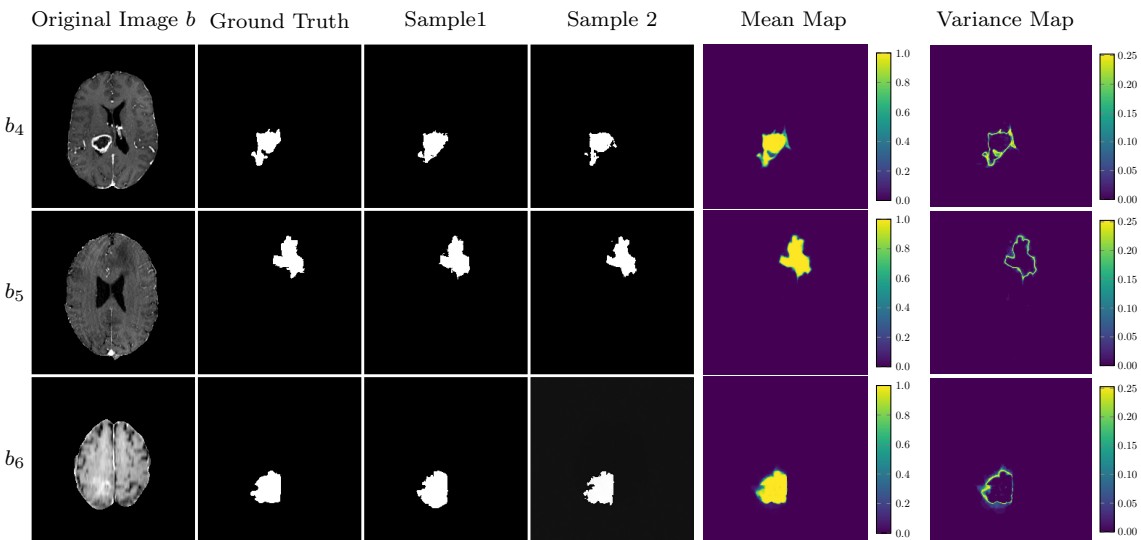

Figure 7: Additional examples of the produced mean and variance maps for 100 sampling runs.

