# OpenReview forum: "Diffusion Models for Implicit Image Segmentation Ensembles"
_MIDL.io/2022/Conference — MIDL 2022_

### Meta-Review · Area_Chair_dFeP · 2022-02-16

**Recommendation:** Accept (Poster)
**Confidence:** 4

**Metareview:**

This work is well-motivated, well-organized and interesting. The source code of implementation is publicly available. The proposed method delivers state-of-the-art performance in segmentation. The uncertainty maps are helpful for interpretation. Two knowledgeable reviewers recommend accept, and one reviewer recommend borderline. Their rebuttal addresses the concerns from all reviewers.

---

### Decision · Program_Chairs · 2022-02-28

Accept